



# The large-scale assessment of avalanche risk for ski resort areas in Northern Caucasus region.

**Komarov Anton Y.[1], Seliverstov Yury G.[1], Glazovskaya Tatyana G.[1], Turchaninova Alla S.[1]**

[1] Lomonosov Moscow State University, Faculty of Geography,
Research Laboratory of Snow Avalanches and Debris Flows,
Moscow, Russian Federation

*Correspondence to:* Komarov A. Y. (ankom9@gmail.com), Glazovskaya T.G. (TG71@yandex.ru)

**ABSTRACT**. Avalanche pose a significant problem in most mountain regions of Russia. The constant growth of economic activity in highlands and the increased avalanche hazard lead to the demand to develop the methods of large-

scale avalanche hazard assessment. This is needed for the determination of appropriate avalanche protection and life safety measures in avalanche-prone areas, as well as for economical reasons. The data obtained from large-scale avalanche risk assessments using our method should be valuable for various economical estimations of developing mountain regions in Russia

The actuality of natural hazard risk estimations is also determined by the Federal Law of Russian Federation. According

to the National Standards, such estimations should take place during the engineering surveys of the object. However, the required standard algorithm and formulas for such assessments do not exist (and cant be found) in official documentation. According this problem, our main purpose was to develop the large-scale risk assessment method and to approbate it on the developing but poorly researched ski resort areas.  This method includes the formulas to calculate collective and individual avalanche risk. The results of risk analysis are shown in quantitative data that can be used to

determine levels of avalanche risk (appropriate, acceptable and inappropriate) and to suggest methods to decrease the individual risk to acceptable level or better. It makes possible to compare risk quantitative data obtained from different regions, analyze it and evaluate the economical feasibility of protection measures.

**KEY WORDS**: *avalanche; risk assessment; hazard; ski resorts; Caucasus; Russia*



## 1 Introduction

Today such researching is essentially important for the territory of the Northern Caucasus. The rapid development of tourism infrastructure take place here due to creation of the number of large ski and tourist resorts. The significantly increased number of visitors is observing in dangerous areas during the last few years. The level of avalanche risk is

growing with equal rates. This activity encourages the development of avalanche risk assessment methods (Seliverstov et al., 2008; Shnyparkov A.L. et al., 2012; Zischg A. et al., 2004, 2005). The governmental standards require more pierce investigations as well (SNIP 11-02-96 update, 2013; SNIP 22-02-2003 update, 2012; Vorob'ev Yu. L., 2005).

The increased number of visitors has been observed since the opening of Rosa Khutor resort in Krasnaya Polyana, Sochi. Some new projected resort areas including Veduchi, Lagonaki and Mamison are at the stage of engineering

survey. Nevertheless, many avalanche-prone areas of Northern Caucasus region are still poorly researched and the lack of avalanche and meteorological data is a usual problem (Myagkov. S. M. et al., 1992). The special avalanche and snow observations are almost absent. The Veduchi (Eastern Caucasus), Lagonaki (Western Caucasus) and Mamison (Central Caucasus) resorts climate and geomorphologic conditions including snow and avalanche characteristics differ significantly (Khrustaleva Y. P., Panova S. V., 2002). That the formula test results should display some notable

deviations for each of this regions as well. The analysis of this information is valuable for further research, creation of avalanche risk classifications and for protection measures development.

### 1.1 Natural conditions

The Veduchi, Mamison and Lagonaki regions are locating in the same mountain system but due to regional heterogeneity of climate circulation and geology the natural conditions including avalanche activity differ considerably

(Atlas, 1997; Zalikhanov M. Ch. (ed), 2004). The dominant western circulation patterns lead to great difference in precipitation and snow accumulation in eastern, central and western regions.

The western region, including Lagonaki area, receive huge amount of snow despite of comparatively small altitudes (up to 2804 m in Lagonaki). The sub-latitudinal Rocky and Side ridges work as the first barrier on the way of air masses movement. The ruggedness of the terrain is quite weak but some very large avalanche catchment zones can be found

in mountain river valleys. The combination of climate and morphology characteristics of this area (Table 1, 2) provide favorable conditions for snow avalanches formation. The considerably small slope angles and strong vegetation are the limiting factors of avalanche activity. Small and medium snow slides and avalanches with high repeatability are most typical for this area.

The eastern regions including Veduchi area is considerably drier. The average precipitation, the duration of snow cover

and the depth of snowpack is much lower then in western region (Table 1). On the other hand, the high altitudes (up to 3021m) and extremely rugged terrain with big slope angles and V-shaped profiles provide necessary conditions for snow avalanche formation (Table 2). This area is characterized by large occasional avalanches with 50+ year return period. Such avalanches may be very destructive due to specific geomorphological conditions of this area. Small avalanches occur almost every year.

The Central Caucasus region includes the Mamison area. The ridges of this area works as the main barrier on the way of moist western air masses, strong precipitation is typical for this highland area. The altitudes exceed 4010 m, this is one of the most high altitude area within Caucasus mountain system. The typical Alpine morphology of the slopes with V-





shaped valley profiles provide favorable conditions for avalanches, as well as great amount of precipitation observing here. The duration of avalanche period, the depth of snowpack and the return period of avalanches are usually higher then in other regions (Table 1). Medium and large avalanches with big volumes, large runout distances and average return periods are most typical in this area (Bolov V. R., Zalikhanov M. Ch., 1984). The climate and geological factors are almost equally important for avalanche activity in this region.

**Table 1.** Climate characteristics

| Climate characteristics | Lagonaki | Mamison | Veduchi |
|---|---|---|---|
| Cyclone frequency | 36% | 37% | 34 % |
| The average and max wind speed M*S$^{-1}$ | 1.5-2 to 35 | 2-8 to 50 | 3-5 to 35 |
| Average january temperature C° | -5 | -10 | -15 |
| The duration of avalanche period Days | 105 | 95 | 80 |
| The average maximum height of snow cover sm | 200 | 150 | 80 |
| The main meteorological factors of avalanching | Heavy snowfall blizzards | Heavy snowfall blizzards recrystallization | Heavy snowfall blizzards |

**Table 2.** Morphology characteristics

| Morphology characteristics | Lagonaki | Mamison | Veduchi |
|---|---|---|---|
| Elevations m | 985 - 2804 м | 1759м - 4018 м | 873 м - 3021м |
| The density of the avalanche catchment zones, sites*km$^{-1}$ | 3-4 | 8 | 5-6 |
| Avalanche return period, years | >10 | >10 | 1-10 |
| The level of avalanche activity | High/Medium | High | High/Medium |





### 1.2 Previous investigations

The Research Laboratory of snow avalanches and debris flows, Geographical faculty, Moscow state University develop the methodology to assess risk and potential natural hazard damage for different scale in order to increase the local population and tourists safety and to protect infrastructure (Seliverstov et al., 2008; Shnyparkov A.L. et al., 2012). The

result of practical applications of these techniques is a large-scale risk zoning of the studied areas of and quantitative values for individual and total social risk. The previous small-scale researches of avalanche risk is Northern Caucasus allowed us to receive some important data about risk distribution in the region. But, due to the economic growth, the more profound investigations of particular objects in large scale become essential. In accordance with previous studies in the laboratory there are three levels of individual risk is "appropriate" (less then $1 \times 10^{-6}$), "acceptable" (up to $1 \times 10^{-4}$)

and "unacceptable" ($1 \times 10^{-4}$). Economic development of the territory must be carried out in accordance with its risk level. We use the same categories for large scale researches.

The first approbation of large-scale avalanche risk estimation methods was performed for the 3 projecting ski resorts with different natural conditions - Veduchi, Lagonaki and Mamison (E, W and Central Caucasus respectively). During the exploration stage of the project we allocated the avalanche catchment zones and analyzed the main characteristics of

avalanche activity for each of the researching regions.

Using certain correlation dependences (Atlas, 1997; Pogorelov A. V., 1998; Pogorelov A. V., 2002) (that are proven and widely used in Russian glaciology) and spatial field data from laboratory expeditions we calculated the snowpack depth values, duration of avalanche-active period, the volume of avalanches for different elevation levels and avalanche return periods for each area. Using the number of calculation values and actual snowpack depth data and the RAMMS

modeling program we simulated the potential avalanche paths with different runout distances and received the avalanche-dynamic characteristics. Calculated values of avalanche activity were used to calculate the avalanche risk for ski resorts.

### 2 Methods

Risk can be described as a multiplication of  probability of a situation and the amount of damage that can be inflicted.

Avalanche risk can be recorded by temporal and spatial overlapping of the two independent processes of avalanche danger and use of the area (Bartelt, P. et al., 2012; Hendrikx J., Owens I. et al., 2006; Seliverstov et al., 2008;  Wilhelm C., 1998).

The use of the area corresponds to the probability of presence and the number of people present. The vulnerability (V) is recorded as a conditional probability under the condition that the avalanche has taken place as well as that the person

was present. In this study we use the extreme values of snowpack which characterize avalanches with 100 year return period.

In order to receive required individual and collective risk for ski resorts, we have defined the following indicators - the spatial (Vs) and temporal (Vt) vulnerability.

The temporal vulnerability of people characterizes the duration of a person staying in an avalanche-prone area. It is

calculated as a function of the duration of human presence ($t_d$ and $t_y$) and its location in a dangerous area (Eq. 1):

$$Vt = td \times ty / (24 \times 365) \qquad (1)$$



The $t_d$ index characterize the average period of a typical representative stay in the targeted object during the day. The $t_y$ index characterize the average period of a typical representative stay in the targeted object during the year. The multiplication of these parameters relatively to the year (24 hours × 365 days) gives us the quantitative values of temporal probability of risk situation.

In this study, we have used the following values: the value of $t_d$ is limited by the duration of chairlifts functioning during the day within the ski complex. This value can vary significantly depending on many factors, but it this study it is averaged to *8 hours* for each resort. The value of $t_y$ is limited by a duration of avalanche period in the study area.

The spatial vulnerability is defined by the exposure of the territory to the impact of snow avalanches. It is calculated as the area of the avalanche-prone territory related to the full area of the polygon (Eq. 2).

$$Vs = Si / S0 \qquad (2)$$

$S_i$ – represents the area of avalanche-prone part of the territory and defined as the total area of the slopes, overlapping by avalanches with 100 year return period (1% probability). $S_0$ – the total area of slopes within the resort.

Full social avalanche risk *(collective risk)* characterizes the expected average number of people killed in avalanches in the year within the study area. Full social risk ($R_n$) was calculated using the following equation (3)

$$Rn = P \times d \times Vt \times Vs \times K \qquad (3)$$

The K and the d indexes characterize the amount of damage that can be done during the risk situation. The *d* is bound to the number of people using the territory – it shows the maximum possible density of sportsmen on the piste. The *K* index represents the mortality coefficient and reflects the long-term statistics of mortality in avalanches. We use the constant value *0.66* for this coefficient (that bounds to the 30% probability to survive in avalanche after being hit). This

value was obtained by analyzing the laboratory materials for the last 20 years for different regions.

The received values of collective (full social) risk $R_n$ can be used to calculate the individual risk *Ri*. This index represents the risk situation related to an individual (single person), the probability of premature death of an individual in the study area. *Ri* is calculated as the ratio of the total social risk to the total number of people (*D*) on pistes during the year (Eq. 4):

$$Ri = Rn / D \qquad (4)$$

The *D* index can vary depending on the temps of resorts development, so we tried different scenarios (50, 150 and 600 thousand visitors per year) for each one. The received information is useful for further resort planing and protection measures development in North Caucasus region.

Territories with individual risk values less then $1\times10^{-6}$ have «*appropriate risk level*». Such territories usually don't need

any avalanche protection measures or special restrictions on the construction of buildings. The values of $1\cdot10^{-6} - 1\times10^{-4}$ characterizes the «*acceptable avalanche risk*». Regions with acceptable risk require specific measures to protect community and infrastructure. The construction is possible here, but appropriate protection measures are highly recommended. If the measures are effective enough it is possible to reduce the coefficient down to appropriate risk level. If the individual risk exceeds $1\times10^{-4}$ the territory have «*unacceptable risk level*». This level characterize

territories with high avalanche activity and rapidly developing infrastructure. Such territories require some urgent measures. The entire spectrum of avalanche protection measures shall be used in order to protect existing facilities and



population and to reduce the risk level. New construction should not be allowed in such territories without special avalanche studies.

**3 Results**

Using these methods we calculated collective and individual risk values for Veduchi, Mamison and Lagonaki resort
areas and analyzed the results. All calculations were performed on the basis of data obtained using MapInfo, ArcGis and RAMMS GIS software.

Full social avalanche risk *(collective risk)* characterizes the expected average number of people killed in avalanches in the year within the study area. Full social risk ($R_n$) was calculated using the following equation (3).

$$Rn = P \times d \times Vt \times Vs , \times K \qquad (3)$$

The meaning of the indexes has already been described in previous paragraph, so we publish only obtained results here in (Table 3).

**Table 3.** Indexes values

| Resort | td | ty | Vt | Vs | d | K | P | Rп |
|--------|-----|-----|-------|------|------|------|------|------|
| Veduchi | 8 | 80 | 0.073 | 0.69 | 4500 | 0.66 | 0.01 | 1,49 |
| Mamison | 8 | 100 | 0.091 | 0.65 | 4500 | 0.66 | 0.01 | 1.75 |
| Lagonaki | 8 | 105 | 0.096 | 0.30 | 4500 | 0.66 | 0.01 | 0,85 |

The td, d, K and P indexes have constant values for all the resorts. The ty, Vt, Vs and Rc indexes vary due to different natural conditions of the regions.

The $t_y$ index characterize the average period of a typical representative stay in the targeted object during the year. It is limited by a duration of avalanche period in the study area and by duration of resorts functioning. For Veduchi, Mamison and Lagonaki resorts it equals 80, 100 and 105 days respectively and limited mostly by avalanche period duration.

The multiplication of ty and td parameters relatively to the year (24 hours × 365 days) gives us the quantitive values of
temporal probability of risk situation Vt. The index values vary from 0.073 in Veduchi to 0.091 in Mamison and 0.096 in Lagonaki.

The area of avalanche catchment zones (So) within the pistes (Si) characterize the Vs index, which represents the ratio of dangerous area related to full (total) area of pistes. Vs index vary from 0.69 in Veduchi (69% of pistes are overlapping with avalanche catchment areas) to 0.65 (65%) in Mamison and 0.30 (30%) in Lagonaki. The calculation of
Vs is a controversial question that requires more precise investigations. It can be refined by inputing decreasing coefficients to the formula in order to estimate the actual area of dangerous zone for each training level depending on sportsmen speed and possibility to escape the avalanche.

Multiplying the indexes values using equation (3) we determined the collective risk Rc values for each region and received the following results. The collective risk values equals 1.49 ppl*km$^2$/year for Veduchi, 1.75 ppl*km$^2$/year for
Mamison and 0.85 ppl*km$^2$/year for Lagonaki regions.

Then, using the equation (4), we estimated the individual risk values. The individual risk Ri represents the risk situation related to an individual (single person), the probability of premature death of an individual in the study area. Ri is



calculated as the ratio of the total social risk to the total number of people (N) on pistes during the year:

$$Ri = Rn / N \qquad (4)$$

The N index can vary significantly depending on the temps of resorts development. For ski resorts which has not yet started to function it is advisable to take into account different scenarios of its development. Assuming that the number

of guests at the initial stage there will be about 50 000 people/year, then will increase to 150,000 people/year and will reach 600,000 people/year we obtained the following values of individual avalanche risk (Table 4).

**Table 4.** Individual risk values for Veduchi, Mamison and Lagonaki ski resorts.

| Number of visitors, ppl | 50000 | 150000 | 600000 |
|---|---|---|---|
| Veduchi | $2.9*10^{-5}$ | $9.9*10^{-6}$ | $2.5*10^{-6}$ |
| Mamison | $3.5*10^{-5}$ | $1.2*10^{-5}$ | $2.9*10^{-6}$ |
| Lagonaki | $1.7*10^{-5}$ | $5.6*10^{-6}$ | $1.4*10^{-6}$ |

All the calculated values correspond to "acceptable" individual risk level. Consequently it will be necessary to take protection measures, in order to decrease the figure to appropriate values, i.e. less than $1*10^{-6}$. These values can be

achieved by applying various primary and secondary protection measures including warning announcements, closing of pistes when the possibility of avalanche situation acquires extreme values, active influence on snow using different methods. Construction of special avalanche protective structures is quite expensive, but often it is the only way to make the territory safe.

**4 Discussion**

The received results allows us to estimate the risk levels for different territories and to suggest the most effective protection measures for ski resorts. These calculations represent quite rough approximations. Each component of the formula can be refined in order to obtain more accurate (precise) results, but require more pierce investigations.
The calculation of d and Vs indexes is the most controversial question so we have analyzed the way how they can be refined.

The d index can vary widely depending on many factors, such as time, season, and spatial distribution of sportsmen on the piste. The spatial distribution shows a good correlation with the training levels of sportsmen. Using the materials (Shealy J. et al., 2005; Williams R. et al., 2007) and official resort statistics we tried to determine appropriate people density and the maximum possible number of people on the piste at the same time for Caucasus ski resorts for 3 professional levels (beginners, medium and professional level gradations). We also have analyzed the percentage ration

of groups with different training level and estimated their average movement speed (Table 5). The average movement speed of sportsmen was determined using the results of (Shealy J., Ettlinger C., Johnson R., 2005) researches. As long as skiers and snowboarders usually move fast while riding the piste and can reach considerable speeds, they can exceed the speed of the avalanche on certain sections of pistes. Consequently people are able to avoid avalanche if they move quick enough while riding one of these sections. For athletes with good training level and high movement speed this




capability is much higher than that of the beginners. Thus the size of the dangerous zone shall be reduced depending on the training level for each group.

Comparing the calculated speeds (using RAMMS software) of avalanches in different parts of trails with average movement speeds of sportsmen, we determined the areas, were the sportsmen speed exceed the speed on an avalanche

5 and estimated the possibilities to avoid an avalanche for each of these group. Comparing this area to the full avalanche-prone area, we can receive the M coefficient that shall be used in spatial vulnerability calculations (Table 7). These clarifications help us to estimate the real number of victims more precisely. The results are shown in tables 5, 6 and 7.

**Table 5.** The d index and average sportsmen movement speed.

| Training level | Maximum appropriate density of sportsmen on the piste ppl/km2 | The average ratio of different training level groups on the piste % | The average number of people according to the ratio ppl/km2 | The average movement speed km/h |
|---|---|---|---|---|
| Professional | 2000 | 15 | 300 | 65 |
| Middle | 4000 | 60 | 2400 | 32 |
| Newbies | 7500 | 25 | 1800 | 16 |
| Average | 4500 | 100 | 4500 | |

**Table 6.** The % of the area where maximum avalanche speed exceed the average movement speed of sportsmen  (16,

10 32 and 65 km/h gradations)

| Territory | Maximum avalanche speed exceed 16 km/h | Maximum avalanche speed exceed  32 km/h | Maximum avalanche speed exceed 65 km/h |
|---|---|---|---|
| Lago Naki | 95% | 90% | 65% |
| Veduchi | 92% | 80% | 58% |
| Mamison | (93%) | (85%) | (60%) |

**Table 7.** The M index for Veduchi resort.

| Training level | The area of dangerous zone compared to the full area of avalanche catchment zone for each professional class. M index - Veduchi |
|---|---|
| Professional | 0.58 |
| Middle | 0.8 |
| Newbies | 0.92 |
| All (according to the ratio) | 0.81 |

The other indexes can be refined by similar ways, but require detailed statistical information which is absent for



selected regions. We believe that such data can be obtained if existing ski resorts will keep statistics of   some parameters (td, ty, d, Vs) used in our formula and climate characteristics.

**5 Conclusion**

The aim of this research was to elaborate a method of avalanche social risk estimation for local objects such as ski

resorts and other rapidly developing mountain areas.

The previously used methodology of small-scale avalanche risk assessment  was modified in order to use in large scale. This methodic shows good results for Caucasus region resorts, but it requires more precise investigations and more accurate statistical information. The improvement of  risk assessment methods  is associated with clarification of such indicators as the number of visitors to the resort, change in the density of tourists on the route at different times of the

day and year, long-term statistical meteorological data (including avalanche activity and snow coverage indicators).

As a result of the performed calculations we established that all the calculated values correspond to "acceptable" individual risk level. Consequently it will be necessary to take protection measures, in order to risk to appropriate values, i.e. less than 1*10-6.  These values can be achieved by applying various primary and secondary protection measures including warning announcements, closing of pistes when the possibility of avalanche situation acquires

extreme values, active influence on snow using different methods. It is necessary to develop interventions in order to determine how the use of different avalanche-protection events will change the risk indicators and recommend the most advantageous solutions.

*Acknowledgements.* The authors would like to thank Shnyparkov A.L. and Sokratov S.A. for valuable information, editorial comments and publication support. We also wish to thank all the collective of the laboratory for technical

assistance.





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
