# Peer review of "The large-scale assessment of avalanche risk for ski resort areas in Northern Caucasus region."

_Natural Hazards and Earth System Sciences, 2016_

## Referee Comment (RC1) · Anonymous Referee #1 · 11 Apr 2016

The paper on risk assessments in ski resorts introduces a way to assess the risk in ski areas. The focus is on the risk on slopes. Reading the abstract rises great interest, but the paper does not hold what the abstract promises. In the last sentence of the introduction (page 2 line 15) the authors state that "the analysis of this information is valuable for further research, creation of avalanche risk classifications and for protection measures development ". I believe that only the first point applies – the work presented is valuable for further research, however, the uncertainties in the applied methods are far too high that the approach needs to be adapted before it can be applied. The greatest shortcoming is that the authors did not define clear objectives and mix different risk assessment approaches. There is lots of literature available and should be used by

the authors to align their research. Anyhow, the idea to calculate the risk for "sports-men" on slopes is worthwhile considering, when discussed appropriately. The main questions in the methodology from my side are: - What is your objective? Once you talk about ski resorts in general for construction purposes, but then you only look at sportsmen on the slopes - Why did you choose a 100 year return period? - How did you consider that some avalanches may reach the slope every year, other only every 50 years? - The discussion on the density of sportsmen on the slope is a bit arbitrary. How did you consider that in time you would expect a 100 year avalanche the weather might be poor? No visibility? Even the good skier cannot ski fast enough to escape the avalanche (which is a factor a bit questionable anyway)? - How did you consider that faster skiers, usually ski more runs per day and in the end may be in total be more exposed than bad skiers? - Are you only looking at natural avalanche activity? More skiers may die in avalanches they trigger by themselves not on a slope, but off-piste - The average avalanche may reach 100 km/h (you look at a 100 year avalanche event) – the 60 km/h you assume for good skier does not help - How do the sportsmen know that he is endangered by an avalanche and has to speed up? - The speed is terrain specific and should be estimated for each avalanche catchment - I believe that you should more consider the individual avalanche path - It would be better for the reader to have more figures and pictures instead of tables - Even though the text is under-standable, the English needs polishing - Why don't you distinguish between the risk to infrastructure and the risk to people? The latter one should also consider the time people spent in e.g. restaurants or in lifts. - Why do you not use more scenarios or sensitivity analysis? - …….. All in all I believe that the paper could be improved by defining clear objectives and based on these a better structuring. For now, the work is a good scientific exercise, however the practical significance is debateable.

---

## Referee Comment (RC2) · Anonymous Referee #2 · 21 Apr 2016

Referee's report on nhess-2016-68, The large-scale assessment of avalanche risk for ski resort areas in Northern Caucasus region

The authors present a regional risk assessment for snow avalanches, focusing on the scale of ski resorts in the Northern Caucasus, Russian Federation. They compute the individual and collective fatality risk based on collected data, estimated amounts of elements at risk (skiers) and avalanche simulation performed with RAMMS. As such, the topic is of considerable interest to the readers of NHESS, and the topic should be considered for publication.

However, there are some shortcomings in the current version of the NHESSD paper which I will address below. These shortcomings should be considered by the authors

before the manuscript may become acceptable for inclusion in NHESS.

The main point of criticism is related to the main message of the paper: if people on ski-runs only would drive fast enough downhill, their individual risk to die in a snow avalanche would decrease. I strongly believe that such messages are against ANY common sense, because this would mean that "if car drivers would only drive fast enough, they will never experience a car accident".

Despite this rugged statement, the method applied seems to be reliable to compute individual and collective risk, but I have some concerns described below in a chronological order.

Abstract

Page 1, line 15: "...should take place during the engineering surveys of the object" -> please specify, this is not clear to me

Intrdoduction

Page 2, line 2: Please start the introduction with more general statements funneling down later to the case study of the Northern Caucasus. IT may be worth to check available literature on tourism infrastructure and risk assessments, and then come to the case study. The statement of increasing visitors during the last years needs a Citation.

Page 2, line 11: "The special avalanche and snow observations are almost absent [in] Veduchi..." -> needs clarification, I do not really understand. Same with "That the formula test results should display some notable deviations for each of the regions as well."

Section 1.1

Here I would like to suggest a map showing the readers the location of the test sites.

Table 1: From the information provided in relation to the cyclone frequency I cannot

[Figure]

deduce the differences between the three case studies mentioned on page 2, line 19 – the frequency is between 34% and 37%.

Section 1.2

This section on previous studies needs clarification, for details please see the attachment. The main points are:

- Where do the numbers for "appropriate", "acceptable" and "unacceptable" risk have their origin?

- Page 4, lines 16-22: This information should go to the Methods section, and needs more specifications beyond "using certain correlation dependencies..."

Section 2

Page 4, lines 24 ff: I would like to suggest a more proper reference to available literature on snow avalanche risk, including the "classical" risk equations (risk = hazard x vulnerability or risk = hazard x elements at risk x vulnerability) and how these approaches rooting in risk evaluation for either settlements or traffic infrastructure were modified in order to meet the requirements to compute individual fatality risk. Moreover, it seems debatable why the 1/100 year avalanche is used during computation – in the European Alps, the more frequent but smaller avalanches seem to be the challenge for individual risk of skiers.

Page 4, lines 28 f: From my point of view, vulnerability to snow avalanches is related to the impact pressure – and then secondary dependent on the characteristics of elements at risk. For skiers, however, the use of impact pressure would be misleading. As such it would be useful to compare the approach presented to a more empirical approach: Wilhelm (1997, p. 76), to give an example, shows the probability of dying in an avalanche by using data collected by the Swiss Federal Institute of Snow and Avalanche Research and he concludes that the probability of death is 32 %. It may also be worth to check the studies by Keiler, Fuchs, Sokratov and Shnyparkov related

to the short-term avalanche risk (Fuchs et al. 2013; Keiler et al. 2005) – I just mention this because some of them are acknowledged in the end of this manuscript.

Equation (1) needs careful interpretation because (1) the avalanche hazard is variable over time and (2) during the summer months there may not be any risk due to the closure of the skiing areas. As such, the 365 days should be re-interpreted.

Equation (2) relies on the assumption that the number of people on a ski slope are homogenously distributed, which may not be the case due to the network of ski runs, the ski lift infrastructure as well as other infrastructure such as shelters and mountain restaurants. This could be addressed also in the discussion section.

Equation 3: The explanation on page 5, line 16 is not clear to me, and needs specification. Moreover, why did you use a 66 % change to die, where does this data come from? The statement on data made in lines 16-20 could be better supported by published material rather than by "analyzing the laboratory material".

Section 3

In general lots of material presented here is a repetition of the methods section. The authors could (and should) focus on the results instead. I am missing a sound description of the materials presented in Tables 3-7. My main concern about speeding the ski-run in order to survive was already introduced above.

Equations (3) and (4) were already presented in the methods section, moreover, Equation (4) in the results section makes use of different indices. The information given on page 7, lines 5-6 is inconsistent to page 5, line 26.

Page 7, line 10: primary and secondary measures needs clarification, as well as the sentence on "active influence on snow using different methods".

Section 4

The statement on page 7, lines 20-22 needs clarification; what do you mean when

writing "the spatial distribution [of what] shows a good correlation with the training levels of sportsmen. Using the materials [which] and official statistics [which are. . .] (. . .)" the density of people was determined? Moreover, I kindly would like to suggest that the authors clearly address the limitations of their study, for example with respect to the homogenous distribution of people on the ski runs, the precision of RAMMS, the temporal aspect that during the weekend and during holidays there may be more people present, etc. It is also worth to think about the general small risk values for the individual snow avalanche risk (Table 4) in comparison to other risks such as traffic accidents in Russia or health risks. From Table 6 it can be concluded that we should prohibit skiing for beginners, intermediate skiers and even professional skiers due to the distribution of snow avalanche velocities – in almost two third of the area the snow avalanche speed exceeds the speed of professional skiers which means that finally 2 out of 3 will not be able to escape.

More remarks are given in the supplement to this review. Further the manuscript needs a proper proof-read by a native speaker, there are some classical errors from the Russian translation (e.g., the missing articles) but also the overall wording should be improved in order to enhance the readability.

References mentioned

Fuchs, S., Keiler, M., Sokratov, S. A., and Shnyparkov, A.: Spatiotemporal dynamics: the need for an innovative approach in mountain hazard risk management, Natural Hazards, 68, 1217-1241, 2013.

Keiler, M., Zischg, A., Fuchs, S., Hama, M., and Stötter, J.: Avalanche related damage potential – changes of persons and mobile values since the mid-twentieth century, case study Galtür, Natural Hazards and Earth System Sciences, 5, 49-58, 2005.

Wilhelm, C.: Wirtschaftlichkeit im Lawinenschutz, Mitteilungen des Eidgenössischen Instituts für Schnee- und Lawinenforschung, Davos, 309 pp., 1997.

Please also note the supplement to this comment:
http://www.nat-hazards-earth-syst-sci-discuss.net/nhess-2016-68/nhess-2016-68-
RC2-supplement.pdf

**Supplement:**

[revised manuscript text omitted]

---

## Referee Comment (RC3) · Anonymous Referee #3 · 1 May 2016

A preliminary assessment of avalanche risk is necessary for designing the new ski resorts. Studies of avalanche danger and avalanche activity on a large scale have to be conducted for this purpose. The problem of avalanche risk assessment for new resorts in Russia has to solve in a lack of information on avalanche activity. Therefore it is necessary to analyze the avalanche formation natural conditions. Thus the problem discussed in the article is interesting and relevant.

Subjects of the article correspond to the direction of the journal. The avalanche formation conditions are studied in the three regions of the Caucasus (in the west, center and east of the ridge). The level of avalanche risk has been calculated by an analysis of natural conditions.

[Figure]

The following comments are on the content of the article.

It is necessary to give a map of the area indicating the places studied, and it is desirable to pictures of the landscapes.

The natural conditions are described not enough detail. There are no quantitative characteristics of the terrain (the depth of the valleys, steep slopes, vegetation type, and etcetera) and avalanche danger (avalanche catchment areas, the number of days with avalanches, avalanche volumes).

The data on sources of climate data are absent. It is necessary to specify the height of the weather stations, their location relative to the studied areas. High-altitude zoning climatic characteristics (temperature, rainfall, snow cover, wind speed) is not considered.

The mortality rate of 0.66 is questionable. It should be lower at ski resorts with the avalanche safety services. It is equal to 0.47 in our region.

The hypothesis that the mortality rate depends on the qualification of skiers is very controversial. In any case, the introduction of the amendment clarifies the risk level by only 20%, which is much less than the errors in determining the other components of the avalanche risk.

It is hardly necessary to calculate the avalanche risk for the entire year. It is more logical to estimate it for the period of the ski season. Instead avalanche duration period (from the first to the last of the avalanche) we must take the number of days with avalanches. None of the skier is located in avalanche danger zone 8 hours per day.

The number of people considered to be in some cases on the resort, in others - on the pistes.

The equations have the symbols d, and D. How are they different? Not all parameters have dimension.

The parameters in Formula 3 have not justification.

Lines 11-15 on p.9 duplicate lines 8-13 on page 7.

The article can be published after considerable improvements.

————————————————

---

## Author Comment (AC1) · 26 Jun 2016

Thank you for your interest to the article. We believe that your comments and remarks are reasonable and valuable, so we will take them into account in order to compile the final version of the article. We agree that the objective of the article is not clear enough, so we will do our best to make it more understandable. The main objective of this article is to develop the large-scale risk assessment method (for people, not for infrastructure) and to approbate it using factual data from new Caucasus ski resorts. We understand that such characteristic as density of people in avalanche-prone area is debatable and depends of many factors (such as time of day, date, weather, economical conditions, policy of resort an so on). Of course each of these factors should be analyzed appropri-

ately in order to increase the accuracy of risk assessment method. However, the main purpose of this research is to create and approbate the basic formula, that may be refined and developed further using additional factors and clarifications. We use 100-year return period for our RAMMS model, the maximum appropriate people density on slope and an assumption that the resort will be open in order to show the maximum risk for such a catastrophic situation. We understand that it may be only regarded as one of the components of risk and shall be refined further. As for the sportsmen\avalanche speed and the possibility to escape the avalanche, we believe that these factors shall only be mentioned in this article as one of the additional parameters, that may be valuable for further clarifications. However, you are absolutely right about the fact that the speed is terrain specific and should be estimated for each individual avalanche path, so we are going to cover this topic in the next article in order to make it more clear for the reader. Avalanches triggered by backcountry sportsmen should be regarded as an important component of risk estimations but require a separate detailed research. We will try to do our best to re-structure an article and to make it more clear for the reader.

PS We also excuse for poor english and ask you to take into consideration that this is the first international article for the author (Komarov A, junior research scientist)

---

## Author Comment (AC2) · 26 Jun 2016

Thank you for your interest to the article. We believe that your comments and remarks are reasonable and valuable, so we will take them into account in order to compile the final version of the article. We agree that the objective of the article is not clear enough, so we will do our best to make it more understandable.

The main objective of this article is to develop the large-scale risk assessment method (for people, not for infrastructure) and to approbate it using factual data from new Caucasus ski resorts. We understand that such characteristic as density of people in avalanche-prone area is debatable and depends of many factors (such as time of day, date, weather, economical conditions, policy of resort an so on). Of course each

of these factors should be analyzed appropriately in order to increase the accuracy of risk assessment method. However, the main purpose of this research is to create and approbate the basic formula, that may be refined and developed further using additional factors and clarifications. We use 100-year return period for our RAMMS model, the maximum appropriate people density on slope and an assumption that the resort will be open in order to show the maximum risk for such a catastrophic situation. We understand that it may only be regarded as one of the components of risk and shall be refined further.

As for the sportsmen\avalanche speed and the possibility to escape the avalanche, we believe that these factors shall only be mentioned in this article as one of the additional parameters, that may be valuable for further clarifications. However, you are absolutely right about the fact that the speed is terrain specific and should be estimated for each individual avalanche path, so we are going to cover this topic in the next article in order to make it more clear for the reader. The question of sportsmen/avalanche speed and possibility to escape the avalanche is very debatable. It should be regarded as an important component of risk estimations but and require a separate detailed research. We will try to do our best to re-structure an article and to make it more clear for the reader. First of all we are going to check each of your remarks and to fix all the inaccuracies including maps, climate characteristics and citations. We are going to review all the formulas and to make some clarifications in accordance to your comments.

PS We also excuse for poor english and ask you to take into consideration that this is the first international article for the author (Komarov A, junior research scientist)
* * *